# Anchor-Free Correlated Topic Modeling: Identifiability and Algorithm

**Kejun Huang**[*]  **Xiao Fu**[*]  **Nicholas D. Sidiropoulos**
Department of Electrical and Computer Engineering
University of Minnesota
Minneapolis, MN 55455, USA
huang663@umn.edu  xfu@umn.edu  nikos@ece.umn.edu

## Abstract

In topic modeling, many algorithms that guarantee identifiability of the topics have been developed under the premise that there exist anchor words – i.e., words that only appear (with positive probability) in one topic. Follow-up work has resorted to three or higher-order statistics of the data corpus to relax the anchor word assumption. Reliable estimates of higher-order statistics are hard to obtain, however, and the identification of topics under those models hinges on uncorrelatedness of the topics, which can be unrealistic. This paper revisits topic modeling based on second-order moments, and proposes an anchor-free topic mining framework. The proposed approach guarantees the identification of the topics under a much milder condition compared to the anchor-word assumption, thereby exhibiting much better robustness in practice. The associated algorithm only involves one eigen-decomposition and a few small linear programs. This makes it easy to implement and scale up to very large problem instances. Experiments using the TDT2 and Reuters-21578 corpus demonstrate that the proposed anchor-free approach exhibits very favorable performance (measured using coherence, similarity count, and clustering accuracy metrics) compared to the prior art.

## 1 Introduction

Given a large collection of text data, e.g., documents, tweets, or Facebook posts, a natural question is what are the prominent topics in these data. Mining topics from a text corpus is motivated by a number of applications, from commercial design, news recommendation, document classification, content summarization, and information retrieval, to national security. *Topic mining*, or *topic modeling*, has attracted significant attention in the broader machine learning and data mining community [1].

In 2003, Blei *et al.* proposed a Latent Dirichlet Allocation (LDA) model for topic mining [2], where the topics are modeled as probability mass functions (PMFs) over a vocabulary and each document is a mixture of the PMFs. Therefore, a word-document text data corpus can be viewed as a matrix factorization model. Under this model, posterior inference-based methods and approximations were proposed [2, 3], but identifiability issues – i.e., whether the matrix factors are unique – were not considered. Identifiability, however, is essential for topic modeling since it prevents the mixing of topics that confounds interpretation.

In recent years, considerable effort has been invested in designing identifiable models and estimation criteria as well as polynomial time solvable algorithms for topic modeling [4, 5, 6, 7, 8, 9, 10, 11]. Essentially, these algorithms are based on the so-called *separable nonnegative matrix factorization* (NMF) model [12]. The key assumption is that every topic has an 'anchor word' that only appears in that particular topic. Based on this assumption, two classes of algorithms are usually employed, namely linear programming based methods [5, 7] and greedy pursuit approaches [11, 6, 8, 10]. The

---

[*]These authors contributed equally.

former class has a serious complexity issue, as it lifts the number of variables to the square of the size of vocabulary (or documents); the latter, although computationally very efficient, usually suffers from error propagation, if at some point one anchor word is incorrectly identified. Furthermore, since all the anchor word-based approaches essentially convert topic identification to the problem of seeking the vertices of a simplex, most of the above algorithms require normalizing each data column (or row) by its $\ell_1$ norm. However, normalization at the factorization stage is usually not desired, since it may destroy the good conditioning of the data matrix brought by pre-processing and amplify noise [8].

Unlike many NMF-based methods that work directly with the word-document data, the approach proposed by Arora *et al.* [9, 10] works with the pairwise word-word correlation matrix, which has the advantage of suppressing sampling noise and also features better scalability. However, [9, 10] did not relax the anchor-word assumption or the need for normalization, and did not explore the symmetric structure of the co-occurrence matrix – i.e., the algorithms in [9, 10] are essentially the same asymmetric separable NMF algorithms as in [4, 6, 8].

The anchor-word assumption is reasonable in some cases, but using models without it is more appealing in more critical scenarios, e.g., when some topics are closely related and many key words overlap. Identifiable models without anchor words have been considered in the literature; e.g., [13, 14, 15] make use of third or higher-order statistics of the data corpus to formulate the topic modeling problem as a tensor factorization problem. There are two major drawbacks with this approach: i) third- or higher-order statistics require a lot more samples for reliable estimation relative to their lower-order counterparts (e.g., second-order word correlation statistics); and ii) identifiability is guaranteed only when the topics are uncorrelated – where a super-symmetric parallel factor analysis (PARAFAC) model can be obtained [13, 14]. Uncorrelatedness is a restrictive assumption [10]. When the topics are correlated, the model becomes a Tucker model which is not identifiable in general; identifiability needs more assumptions, e.g., sparsity of topic PMFs [15].

**Contributions.** In this work, our interest lies in topic mining using word-word correlation matrices like in [9, 10], because of its potential scalability and noise robustness. We propose an anchor-free identifiable model and a practically implementable companion algorithm. Our contributions are two-fold: First, we propose an anchor-free topic identification criterion. The criterion aims at factoring the word-word correlation matrix using a word-topic PMF matrix and a topic-topic correlation matrix via minimizing the determinant of the topic-topic correlation matrix. We show that under a so-called *sufficiently scattered* condition, which is much milder than the anchor-word assumption, the two matrices can be uniquely identified by the proposed criterion. We emphasize that the proposed approach does not need to resort to higher-order statistics tensors to ensure topic identifiability, and it can naturally deal with correlated topics, unlike what was previously available in topic modeling, to the best of our knowledge. Second, we propose a simple procedure for handling the proposed criterion that only involves eigen-decomposition of a large but sparse matrix, plus a few small linear programs – therefore highly scalable and well-suited for topic mining. Unlike greedy pursuit-based algorithms, the proposed algorithm does not involve deflation and is thus free from error propagation; it also does not require normalization of the data columns / rows. Carefully designed experiments using the TDT2 and Reuters text corpora showcase the effectiveness of the proposed approach.

## 2 Background

Consider a document corpus $\boldsymbol{D} \in \mathbb{R}^{V \times D}$, where each column of $\boldsymbol{D}$ corresponds to a document and $\boldsymbol{D}(v, d)$ denotes a certain measurement of word $v$ in document $d$, e.g., the word-frequency of term $v$ in document $d$ or the term frequency–inverse document frequency (tf-idf) measurement that is often used in topic mining. A commonly used model is

$$\boldsymbol{D} \approx \boldsymbol{C}\boldsymbol{W}, \tag{1}$$

where $\boldsymbol{C} \in \mathbb{R}^{V \times F}$ is the word-topic matrix, whose $f$-th column $\boldsymbol{C}(:, f)$ represents the probability mass function (PMF) of topic $f$ over a vocabulary of words, and $\boldsymbol{W}(f, d)$ denotes the weight of topic $f$ in document $d$ [2, 13, 10]. Since matrix $\boldsymbol{C}$ and $\boldsymbol{W}$ are both nonnegative, (1) becomes a nonnegative matrix factorization (NMF) model – and many early works tried to use NMF and variants to deal with this problem [16]. However, NMF does not admit a unique solution in general, unless both $\boldsymbol{C}$ and $\boldsymbol{W}$ satisfy some sparsity-related conditions [17]. In recent years, much effort has been put in devising polynomial time solvable algorithms for NMF models that admit unique factorization. Such models and algorithms usually rely on an assumption called "separability" in the NMF literature [12]:

**Assumption 1** *(Separability / Anchor-Word Assumption) There exists a set of indices* $\Lambda = \{v_1, \ldots, v_F\}$ *such that* $\boldsymbol{C}(\Lambda, :) = \text{Diag}(\boldsymbol{c})$*, where* $\boldsymbol{c} \in \mathbb{R}^F$*.*

In topic modeling, it turns out that the separability condition has a nice physical interpretation, i.e., every topic $f$ for $f = 1, \ldots, F$ has a 'special' word that has nonzero probability of appearing in topic $f$ and zero probability of appearing in other topics. These words are called 'anchor words' in the topic modeling literature. Under Assumption 1, the task of matrix factorization boils down to finding these anchor words $v_1, \ldots, v_F$ since $\boldsymbol{D}(\Lambda, :) = \mathrm{Diag}(\boldsymbol{c})\boldsymbol{W}$ — which is already a scaled version of $\boldsymbol{W}$ — and then $\boldsymbol{C}$ can be estimated via (constrained) least squares.

---

**Algorithm 1:**

Successive Projection Algorithm [6]

---

**input** : $\boldsymbol{D}; F$.
$\boldsymbol{\Sigma} = \mathbf{1}^T \boldsymbol{D}^T$
$\boldsymbol{X} = \boldsymbol{D}^T \boldsymbol{\Sigma}^{-1}$ (normalization);
$\Lambda = \emptyset$;
**for** $f = 1, \ldots, F$ **do**
    $\hat{v}_f \leftarrow \arg\max_{v \in \{1, \ldots, V\}} \|\boldsymbol{X}(:, v)\|_2$;
    $\Lambda \leftarrow [\Lambda, \hat{v}_f]$;
    $\boldsymbol{\Theta} \leftarrow \arg\min_{\boldsymbol{\Theta}} \|\boldsymbol{X} - \boldsymbol{X}(:, \Lambda)\boldsymbol{\Theta}\|_F^2$;
    $\boldsymbol{X} \leftarrow \boldsymbol{X} - \boldsymbol{X}(:, \Lambda)\boldsymbol{\Theta}$;
**end**
**output**: $\Lambda$

---

Many algorithms have been proposed to tackle this index-picking problem in the context of separable NMF, hyperspectral unmixing, and text mining. The arguably simplest algorithm is the so-called successive projection algorithm (SPA) [6] that is presented in Algorithm 1. SPA-like algorithms first define a normalized matrix $\boldsymbol{X} = \boldsymbol{D}^T \boldsymbol{\Sigma}^{-1}$ where $\boldsymbol{\Sigma} = \mathrm{Diag}(\mathbf{1}^T \boldsymbol{D}^T)$ [11]. Note that $\boldsymbol{X} = \boldsymbol{G}\boldsymbol{S}$ where $\boldsymbol{G}(:, f) = \boldsymbol{W}^T(f, :)/\|\boldsymbol{W}(f, :)\|_1$ and $\boldsymbol{S}(f, v) = \frac{\boldsymbol{C}(v, f)\|\boldsymbol{W}(f, :)\|_1}{\|\boldsymbol{C}(v, :)\|_1 \|\boldsymbol{D}(v, :)\|_1}$. Consequently, we have $\mathbf{1}^T \boldsymbol{S} = \mathbf{1}^T$ if $\boldsymbol{W} \geq \mathbf{0}$, meaning the columns of $\boldsymbol{X}$ all lie on the simplex spanned by the columns of $\boldsymbol{G}$, and the vertices of the simplex correspond to the anchor words. Also, the columns of $\boldsymbol{S}$ all live in the unit simplex. After normalization, SPA sequentially identifies the vertices of the data simplex, in conjunction with a deflation procedure. The algorithms in [8, 10, 11] can also be considered variants of SPA, with different deflation procedures and pre-/post-processing. In particular, the algorithm in [8] avoids normalization — for real-word data, normalization at the factorization stage may amplify noise and damage the good conditioning of the data matrix brought by pre-processing, e.g., the tf-idf procedure [8]. To pick out vertices, there are also algorithms using linear programming and sparse optimization [7, 5], but these have serious scalability issues and thus are less appealing.

In practice $\boldsymbol{D}$ may contain considerable noise, and this has been noted in the literature. In [9, 10, 14, 15], the authors proposed to use second and higher-order statistics for topic mining. Particularly, Arora *et al.* [9, 10] proposed to work with the following matrix:

$$\boldsymbol{P} = \mathbb{E}\{\boldsymbol{D}\boldsymbol{D}^T\} = \boldsymbol{C}\boldsymbol{E}\boldsymbol{C}^T, \tag{2}$$

where $\boldsymbol{E} = \mathbb{E}\{\boldsymbol{W}\boldsymbol{W}^T\}$ can be interpreted as a topic-topic correlation matrix. The matrix $\boldsymbol{P}$ is by definition a word-word correlation matrix, but also has a nice interpretation: if $\boldsymbol{D}(v, d)$ denotes the frequency of word $v$ occurring in document $d$, $\boldsymbol{P}(i, j)$ is the likelihood that term $i$ and $j$ co-occur in a document [9, 10]. There are two advantages in using $\boldsymbol{P}$: i) if there is zero-mean white noise, it will be significantly suppressed through the averaging process; and ii) the size of $\boldsymbol{P}$ does not grow with the size of the data if the vocabulary is fixed. The latter is a desired property when the number of documents is very large, and we pick a (possibly limited but) manageable vocabulary to work with. Problems with similar structure to that of $\boldsymbol{P}$ also arise in the context of graph models, where communities and correlations appear as the underlying factors. The algorithm proposed in [10] also makes use of Assumption 1 and is conceptually close to Algorithm 1. The work in [13, 14, 15] relaxed the anchor-word assumption. The methods there make use of three or higher-order statistics, e.g., $\underline{\boldsymbol{P}} \in \mathbb{R}^{V \times V \times V}$ whose $(i, j, k)$th entry represents the co-occurrence of three terms. The work in [13, 14] showed that $\underline{\boldsymbol{P}}$ is a tensor satisfying the parallel factor analysis (PARAFAC) model and thus $\boldsymbol{C}$ is uniquely identifiable, *if the topics are uncorrelated*, which is a restrictive assumption (a counter example would be politics and economy). When the topics are correlated, additional assumptions like sparsity are needed to restore identifiability [15]. Another important concern is that reliable estimates of higher-order statistics require much larger data sizes, and tensor decomposition is computationally cumbersome as well.

**Remark 1** *Among all the aforementioned methods, the deflation-based methods are seemingly more efficient. However, if the deflation procedure in Algorithm 1 (the update of $\boldsymbol{\Theta}$) has constraints like in [8, 11], there is a serious complexity issue: solving a constrained least squares problem with $FV$ variables is not an easy task. Data sparsity is destroyed after the first deflation step, and thus even first-order methods or coordinate descent as in [8, 11] do not really help. This point will be exemplified in our experiments.*

## 3  Anchor-Free Identifiable Topic Mining

In this work, we are primarily interested in mining topics from the matrix $\boldsymbol{P}$ because of its noise robustness and scalability. We will formulate topic modeling as an optimization problem, and show

that the word-topic matrix $C$ can be identified under a much more relaxed condition, which includes the relatively strict anchor-word assumption as a special case.

## 3.1 Problem Formulation

Let us begin with the model $P = CEC^T$, subject to the constraint that each column of $C$ represents the PMF of words appearing in a specific topic, such that $C^T \mathbf{1} = \mathbf{1}, \; C \geq \mathbf{0}$. Such a symmetric matrix decomposition is in general not identifiable, as we can always pick a non-singular matrix $A \in \mathbb{R}^{F \times F}$ such that $A^T \mathbf{1} = \mathbf{1}, \; A \geq \mathbf{0}$, and define $\tilde{C} = CA, \; \tilde{E} = A^{-1}CA^{-1}$, and then $P = \tilde{C}\tilde{E}\tilde{C}^T$ with $\tilde{C}^T \mathbf{1} = \mathbf{1}, \; \tilde{C} \geq \mathbf{0}$. We wish to find an identification criterion such that under some mild conditions the corresponding solution can only be the ground-truth $E$ and $C$ up to some trivial ambiguities such as a common column permutation. To this end, we propose the following criterion:

$$\underset{E \in \mathbb{R}^{F \times F}, C \in \mathbb{R}^{V \times F}}{\text{minimize}} \; |\det E|, \quad \text{subject to } P = CEC^T, C^T \mathbf{1} = \mathbf{1}, C \geq \mathbf{0}. \tag{3}$$

The first observation is that if the anchor-word assumption is satisfied, the optimal solutions of the above identification criterion are the ground-truth $C$ and $E$ and their column-permuted versions. Formally, we show that:

**Proposition 1** *Let $(C_\star, E_\star)$ be an optimal solution of (3). If the separability / anchor-word assumption (cf. Assumption 1) is satisfied and $\mathrm{rank}(P) = F$, then $C_\star = C\Pi$ and $E_\star = \Pi^T E\Pi$, where $\Pi$ is a permutation matrix.*

The proof of Proposition 1 can be found in the supplementary material. Proposition 1 is merely a 'sanity check' of the identification criterion in (3): It shows that the criterion is at least a sound one under the anchor-word assumption. Note that, when the anchor-word assumption is satisfied, SPA-type algorithms are in fact preferable over the identification criterion in (3), due to their simplicity. The point of the non-convex formulation in (3) is that it can guarantee identifiability of $C$ and $E$ even when the anchor-word assumption is *grossly* violated. To explain, we will need the following.

**Assumption 2** *(sufficiently scattered) Let $\mathrm{cone}(C^T)^*$ denote the polyhedral cone $\{x : Cx \geq 0\}$, and $\mathcal{K}$ denote the second-order cone $\{x : \|x\|_2 \leq \mathbf{1}^T x\}$. Matrix $C$ is called* **sufficiently scattered** *if it satisfies that: (i) $\mathrm{cone}(C^T)^* \subseteq \mathcal{K}$, and (ii) $\mathrm{cone}(C^T)^* \cap \mathrm{bd}\mathcal{K} = \{\lambda e_f : \lambda \geq 0, f = 1, \ldots, F\}$, where $\mathrm{bd}\mathcal{K}$ denotes the boundary of $\mathcal{K}$, i.e., $\mathrm{bd}\mathcal{K} = \{x : \|x\|_2 = \mathbf{1}^T x\}$.*

Our main result is based on this assumption, whose first consequence is as follows:

**Lemma 1** *If $C \in \mathbb{R}^{V \times F}$ is sufficiently scattered, then $\mathrm{rank}(C) = F$. In addition, given $\mathrm{rank}(P) = F$, any feasible solution $\tilde{E} \in \mathbb{R}^{F \times F}$ of Problem (3) has full rank and thus $|\det \tilde{E}| > 0$.*

Lemma 1 ensures that any feasible solution pair $(\tilde{C}, \tilde{E})$ of Problem (3) has full rank $F$ when the ground-truth $C$ is sufficiently scattered, which is important from the optimization perspective – otherwise $|\det \tilde{E}|$ can always be zero which is a trivial optimal solution of (3). Based on Lemma 1, we further show that:

**Theorem 1** *Let $(C_\star, E_\star)$ be an optimal solution of (3). If the ground truth $C$ is sufficiently scattered (cf. Assumption 2) and $\mathrm{rank}(P) = F$, then $C_\star = C\Pi$ and $E_\star = \Pi^T E\Pi$, where $\Pi$ is a permutation matrix.*

The proof of Theorem 1 is relegated to the supplementary material. In words, for a sufficiently scattered $C$ and an arbitrary square matrix $E$, given $P = CEC^T$, $C$ and $E$ can be identified up to permutation via solving (3). To understand the sufficiently scattered condition and Theorem 2, it is better to look at the dual cones. The notation $\mathrm{cone}(C^T)^* = \{x : Cx \geq 0\}$ comes from the fact that it is the dual cone of the conic hull of the row vectors of $C$, i.e., $\mathrm{cone}(C^T) = \{C^T \theta : \theta \geq 0\}$. A useful property of dual cone is that for two convex cones, if $\mathcal{K}_1 \subseteq \mathcal{K}_2$, then $\mathcal{K}_2^* \subseteq \mathcal{K}_1^*$, which means the first requirement of Assumption 2 is equivalent to

$$\mathcal{K}^* \subseteq \mathrm{cone}(C^T). \tag{4}$$

Note that the dual cone of $\mathcal{K}$ is another second-order cone [12], i.e., $\mathcal{K}^* = \{x | x^T \mathbf{1} \geq \sqrt{F-1}\|x\|_2\}$, which is tangent to and contained in the nonnegative orthant. Eq. (4) and the definition of $\mathcal{K}^*$ in

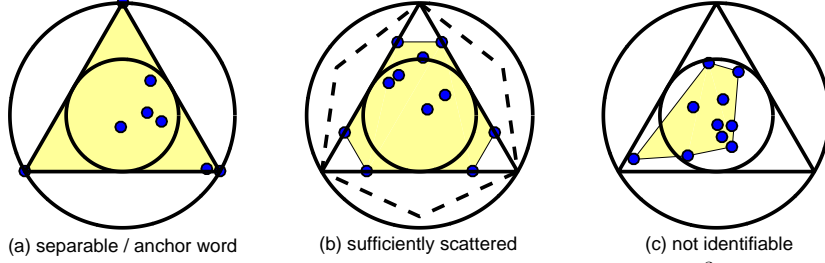

|  (a) separable / anchor word | (b) sufficiently scattered | (c) not identifiable |

Figure 1: A graphical view of rows of $C$ (blue dots) and various cones in $\mathbb{R}^3$, sliced at the plane $\mathbf{1}^T x = 1$. The triangle indicates the non-negative orthant, the enclosing circle is $\mathcal{K}$, and the smaller circle is $\mathcal{K}^*$. The shaded region is $\mathrm{cone}(C^T)$, and the polygon with dashed sides is $\mathrm{cone}(C^T)^*$. The matrix $C$ can be identified up to column permutation in the left two cases, and clearly separability is more restrictive than (and a special case of) sufficiently scattered.

fact give a straightforward comparison between the proposed sufficiently scattered condition and the existing anchor-word assumption. An illustration of Assumptions 1 and 2 is shown in Fig. 1 (a)-(b) using an $F = 3$ case, where one can see that sufficiently scattered is much more relaxed compared to the anchor-word assumption: if the rows of the word-topic matrix $C$ are geometrically scattered enough so that $\mathrm{cone}(C^T)$ contains the inner circle (i.e., the second-order cone $\mathcal{K}^*$), then the identifiability of the criterion in (3) is guaranteed. However, the anchor-word assumption requires that $\mathrm{cone}(C^T)$ fulfills the entire triangle, i.e., the nonnegative orthant, which is far more restrictive. Fig. 1(c) shows a case where rows of $C$ are not "well scattered" in the non-negative orthant, and indeed such a matrix $C$ cannot be identified via solving (3).

**Remark 2** *A salient feature of the criterion in (3) is that it does not need to normalize the data columns to a simplex — all the arguments in Theorem 1 are cone-based. The upshot is clear: there is no risk of amplifying noise or changing the conditioning of $P$ at the factorization stage. Furthermore, matrix $E$ can be any symmetric matrix; it can contain negative values, which may cover more applications beyond topic modeling where $E$ is always nonnegative and positive semidefinite. This shows the surprising effectiveness of the sufficiently scattered condition.*

The sufficiently scattered assumption appeared in identifiability proofs of several matrix factorization models [17, 18, 19] with different identification criteria. Huang *et al.* [17] used this condition to show the identifiability of plain NMF, while Fu *et al.* [19] related the sufficiently scattered condition to the so-called *volume-minimization* criterion for blind source separation. Note that volume minimization also minimizes a determinant-related cost function. Like the SPA-type algorithms, volume minimization works with data that live in a simplex, therefore applying it still requires data normalization, which is not desired in practice. Theorem 1 can be considered as a more natural application of the sufficiently scattered condition to co-occurrence/correlation based topic modeling, which explores the symmetry of the model and avoids normalization.

### 3.2 AnchorFree: A Simple and Scalable Algorithm

The identification criterion in (3) imposes an interesting yet challenging optimization problem. One way to tackle it is to consider the following approximation:

$$\underset{E,C}{\text{minimize}} \ \left\| P - CEC^T \right\|_F^2 + \mu|\det E|, \ \text{subject to} \ C \geq 0, \ C^T\mathbf{1} = \mathbf{1}, \tag{5}$$

where $\mu \geq 0$ balances the data fidelity and the minimal determinant criterion. The difficulty is that the term $CEC^T$ makes the problem tri-linear and not easily decoupled. Plus, tuning a good $\mu$ may also be difficult. In this work, we propose an easier procedure of handling the determinant-minimization problem in (3), which is summarized in Algorithm 2, and referred to as AnchorFree. To explain the procedure, first notice that $P$ is symmetric and positive semidefinite. Therefore, one can apply square root decomposition to $P = BB^T$, where $B \in \mathbb{R}^{V \times F}$. We can take advantage of well-established tools for eigen-decomposition of sparse matrices, and there is widely available software that can compute this very efficiently. Now, we have $B = CE^{1/2}Q, Q^TQ = QQ^T = I$, and $E = E^{1/2}E^{1/2}$; i.e., the representing coefficients of $CE^{1/2}$ in the range space of $B$ must be orthonormal because of the symmetry of $P$. We also notice that

$$\underset{E,C,Q}{\text{minimize}} \ |\det E^{1/2}Q|, \ \text{subject to} \ B = CE^{1/2}Q, \ C^T\mathbf{1} = \mathbf{1}, \ C \geq 0, \ Q^TQ = I, \tag{6}$$

has the same optimal solutions as (3). Since $\boldsymbol{Q}$ is unitary, it does not affect the determinant, so we further let $\boldsymbol{M} = \boldsymbol{Q}^T \boldsymbol{E}^{-1/2}$ and obtain the following optimization problem

$$\underset{\boldsymbol{M}}{\text{maximize}} \ |\det \boldsymbol{M}|, \ \text{subject to } \boldsymbol{M}^T \boldsymbol{B}^T \mathbf{1} = \mathbf{1}, \boldsymbol{B}\boldsymbol{M} \geq 0. \tag{7}$$

By our reformulation, $\boldsymbol{C}$ has been marginalized and we have only $F^2$ variables left, which is significantly smaller compared to the variable size of the original problem $VF + F^2$, where $V$ is the vocabulary size. Problem (7) is still non-convex, but can be handled very efficiently. Here, we propose to employ the solver proposed in [18], where the same subproblem (7) was used to solve a dynamical system identification problem. The idea is to apply the co-factor expansion to deal with the determinant objective function, first proposed in the context of non-negative blind source separation [20]: if we fix all the columns of $\boldsymbol{M}$ except the $f$th one, $\det \boldsymbol{M}$ becomes a linear function with respect to $\boldsymbol{M}(:, f)$, i.e., $\det \boldsymbol{M} = \sum_{k=1}^{F} (-1)^{f+k} \boldsymbol{M}(k, f) \det \bar{\boldsymbol{M}}_{k,f} = \boldsymbol{a}^T \boldsymbol{M}(:, f)$, where $\boldsymbol{a} = [a_1, \ldots, a_F]^T$, $a_k = (-1)^{f+k} \det \bar{\boldsymbol{M}}_{k,f}$, $\forall k = 1, ..., F$, and $\bar{\boldsymbol{M}}_{k,f}$ is a matrix obtained by removing the $k$th row and $f$th column of $\boldsymbol{M}$. Maximizing $|\boldsymbol{a}^T \boldsymbol{x}|$ subject to linear constraints is still a non-convex problem, but we can solve it via maximizing both $\boldsymbol{a}^T \boldsymbol{x}$ and $-\boldsymbol{a}^T \boldsymbol{x}$, followed by picking the solution that gives larger absolute objective. Then, cyclically updating the columns of $\boldsymbol{M}$ results in an alternating optimization (AO) algorithm. The algorithm is computationally lightweight: each linear program only involves $F$ variables, leading to a worst-case complexity of $\mathcal{O}(F^{3.5})$ flops even when the interior-point method is employed, and empirically it takes 5 or less AO iterations to converge. In the supplementary material, simulations on synthetic data are given, showing that Algorithm 2 can indeed recover the ground truth matrix $\boldsymbol{C}$ and $\boldsymbol{E}$ even when matrix $\boldsymbol{C}$ grossly violates the separability / anchor-word assumption.

---

**Algorithm 2:** AnchorFree

**input** : $\boldsymbol{D}$, $F$.
$\boldsymbol{P} \leftarrow$ Co-Occurrence$(\boldsymbol{D})$;
$\boldsymbol{P} = \boldsymbol{B}\boldsymbol{B}^T$, $\boldsymbol{M} \leftarrow \boldsymbol{I}$;
**repeat**
    **for** $f = 1, \ldots, F$ **do**
        $a_k = (-1)^{f+k} \det \bar{\boldsymbol{M}}_{k,f}$, $\forall k = 1, ..., F$;
        // remove $k$-th row and $f$-th column of $\boldsymbol{M}$ to obtain $\bar{\boldsymbol{M}}_{k,f}$
        $\boldsymbol{m}_{\max} = \arg\max_{\boldsymbol{x}} \boldsymbol{a}^T \boldsymbol{x}$ s.t. $\boldsymbol{B}\boldsymbol{x} \geq 0$, $\mathbf{1}^T \boldsymbol{B}\boldsymbol{x} = 1$;
        $\boldsymbol{m}_{\min} = \arg\min_{\boldsymbol{x}} \boldsymbol{a}^T \boldsymbol{x}$ s.t. $\boldsymbol{B}\boldsymbol{x} \geq 0$, $\mathbf{1}^T \boldsymbol{B}\boldsymbol{x} = 1$;
        $\boldsymbol{M}(:, f) = \arg\max_{\boldsymbol{m}_{\max}, \boldsymbol{m}_{\min}} (|\boldsymbol{a}^T \boldsymbol{m}_{\max}|, |\boldsymbol{a}^T \boldsymbol{m}_{\min}|)$;
    **end**
**until** *convergence*;
$\boldsymbol{C}_\star = \boldsymbol{B}\boldsymbol{M}$;
$\boldsymbol{E}_\star = (\boldsymbol{C}_\star^T \boldsymbol{C}_\star)^{-1} \boldsymbol{C}_\star^T \boldsymbol{P} \boldsymbol{C}_\star (\boldsymbol{C}_\star^T \boldsymbol{C}_\star)^{-1}$;
**output**: $\boldsymbol{C}_\star$, $\boldsymbol{E}_\star$

---

## 4 Experiments

**Data** In this section, we apply the proposed algorithm and the baselines to two popular text mining datasets, namely, the NIST Topic Detection and Tracking (TDT2) and the Reuters-21578 corpora. We use a subset of the TDT2 corpus consisting of 9,394 documents which are single-category articles belonging to the largest 30 categories. The Reuters-21578 corpus is the ModApte version where 8,293 single-category documents are kept. The original vocabulary sizes of the TDT2 and the Reuters dataset are $36,771$ and $18,933$, respectively, and stop words are removed for each trial of the experiments. We use the standard tf-idf data as the $\boldsymbol{D}$ matrix, and estimate the correlation matrix using the biased estimator suggested in [9]. A standard pre-processing technique, namely, normalized-cut weighted (NCW) [21], is applied to $\boldsymbol{D}$; NCW is a well-known trick for handling the unbalanced-cluster-size problem. For each trial of our experiment, we randomly draw $F$ categories of documents, form the $\boldsymbol{P}$ matrix, and apply the proposed algorithm and the baselines.

**Baselines** We employ several popular anchor word-based algorithms as baselines. Specifically, the successive projection algorithm (SPA) [6], the successive nonnegative projection algorithm (SNPA) [11], the XRAY algorithm [8], and the fast anchor words (FastAnchor) [10] algorithm. Since we are interested in word-word correlation/co-occurrence based mining, all the algorithms are

Table 1: Experiment results on the TDT2 corpus.

| F | Coh | | | | | SimCount | | | | | ClustAcc | | | | |
|---|---|---|---|---|---|---|---|---|---|---|---|---|---|---|---|
| | FastAchor | SPA | SNPA | XRAY | AnchorFree | FastAchor | SPA | SNPA | XRAY | AnchorFree | FastAchor | SPA | SNPA | XRAY | AnchorFree |
| 3 | -612.72 | -613.43 | -613.43 | -597.16 | **-433.87** | 7.98 | 7.98 | 7.98 | 8.94 | **1.84** | 0.71 | 0.74 | 0.75 | 0.73 | **0.98** |
| 4 | -648.20 | -648.04 | -648.04 | -657.51 | **-430.07** | 10.60 | 11.18 | 11.18 | 13.70 | **2.88** | 0.70 | 0.69 | 0.69 | 0.69 | **0.94** |
| 5 | -641.79 | -643.91 | -643.91 | -665.20 | **-405.19** | 13.06 | 13.36 | 13.36 | 22.56 | **4.40** | 0.63 | 0.63 | 0.62 | 0.64 | **0.92** |
| 6 | -654.18 | -645.68 | -645.68 | -674.30 | **-432.96** | 18.94 | 18.10 | 18.10 | 31.56 | **7.18** | 0.65 | 0.58 | 0.59 | 0.60 | **0.91** |
| 7 | -668.92 | -665.55 | -665.55 | -664.38 | **-397.77** | 20.14 | 18.84 | 18.84 | 39.06 | **4.48** | 0.62 | 0.60 | 0.59 | 0.58 | **0.90** |
| 8 | -681.35 | -674.45 | -674.45 | -657.78 | **-450.63** | 24.82 | 25.14 | 25.14 | 40.30 | **9.12** | 0.57 | 0.56 | 0.58 | 0.57 | **0.87** |
| 9 | -688.54 | -671.81 | -671.81 | -690.39 | **-416.44** | 27.50 | 29.10 | 29.10 | 53.68 | **9.70** | 0.61 | 0.58 | 0.58 | 0.53 | **0.86** |
| 10 | -732.39 | -724.64 | -724.64 | -698.59 | **-421.25** | 31.08 | 29.86 | 29.86 | 53.16 | **13.02** | 0.59 | 0.55 | 0.54 | 0.49 | **0.85** |
| 15 | -734.13 | -730.19 | -730.19 | -773.17 | **-445.30** | 51.62 | 52.62 | 52.62 | 59.96 | **41.88** | 0.51 | 0.50 | 0.50 | 0.42 | **0.80** |
| 20 | -756.90 | -747.99 | -747.99 | -819.36 | **-461.64** | 66.26 | **65.00** | **65.00** | 82.92 | 79.60 | 0.47 | 0.47 | 0.47 | 0.38 | **0.77** |
| 25 | -792.92 | -792.29 | -792.29 | -876.28 | **-473.95** | 69.46 | **66.00** | **66.00** | 101.52 | 133.42 | 0.46 | 0.47 | 0.47 | 0.37 | **0.74** |

Table 2: Experiment results on the Reuters-21578 corpus.

| F | Coh | | | | | SimCount | | | | | ClustAc | | | | |
|---|---|---|---|---|---|---|---|---|---|---|---|---|---|---|---|
| | FastAchor | SPA | SNPA | XRAY | AnchorFree | FastAchor | SPA | SNPA | XRAY | AnchorFree | FastAchor | SPA | SNPA | XRAY | AnchorFree |
| 3 | -652.67 | -647.28 | -647.28 | **-574.72** | -830.24 | 10.98 | 11.02 | 11.02 | **3.86** | 7.36 | 0.66 | 0.69 | 0.69 | 0.66 | **0.79** |
| 4 | -633.69 | -637.89 | -637.89 | **-586.41** | -741.35 | 16.74 | 16.92 | 16.92 | **9.92** | 12.66 | 0.51 | 0.61 | 0.61 | 0.60 | **0.73** |
| 5 | -650.49 | -652.53 | -652.53 | **-581.73** | -762.64 | 21.74 | 21.66 | 21.66 | **13.06** | 15.48 | 0.51 | 0.55 | 0.55 | 0.52 | **0.65** |
| 6 | -654.74 | -644.34 | -644.34 | **-586.00** | -705.60 | 39.9 | 39.54 | 39.54 | 27.42 | **19.98** | 0.47 | 0.49 | 0.50 | 0.46 | **0.64** |
| 7 | -733.73 | -732.01 | -732.01 | **-612.97** | -692.12 | 47.02 | 45.24 | 45.24 | 34.64 | **35.62** | 0.43 | 0.57 | 0.57 | 0.54 | **0.65** |
| 8 | -735.23 | -738.54 | -738.54 | **-616.32** | -726.37 | 85.04 | 83.86 | 83.86 | 82.52 | **62.02** | 0.40 | 0.53 | 0.54 | 0.47 | **0.61** |
| 9 | -761.27 | -755.46 | -755.46 | **-640.36** | -713.81 | 117.48 | 118.98 | 118.98 | 119.28 | **72.38** | 0.37 | 0.56 | 0.56 | 0.47 | **0.59** |
| 10 | -764.18 | -759.40 | -759.40 | **-656.71** | -709.48 | 119.54 | 121.74 | 121.74 | 130.82 | **86.02** | 0.35 | 0.52 | 0.52 | 0.42 | **0.59** |
| 15 | -800.51 | -801.17 | -801.17 | **-585.18** | -688.39 | 307.86 | 309.7 | 309.7 | 227.02 | **124.6** | 0.33 | 0.40 | 0.40 | 0.42 | **0.53** |
| 20 | -859.48 | -860.70 | -860.70 | **-615.62** | -683.64 | 539.58 | 538.54 | 538.54 | 502.82 | **225.6** | 0.31 | 0.36 | 0.36 | 0.38 | **0.52** |
| 25 | -889.55 | -890.16 | -890.16 | **-633.75** | -672.44 | 674.78 | 673 | 673 | 650.96 | **335.24** | 0.26 | 0.33 | 0.32 | 0.37 | **0.47** |

combined with the framework provided in [10] and the efficient `RecoverL2` process is employed for estimating the topics after the anchors are identified.

**Evaluation** To evaluate the results, we employ several metrics. First, *coherence* (`Coh`) is used to measure the single-topic quality. For a set of words $\mathcal{V}$, the coherence is defined as $\text{Coh} = \sum_{v_1, v_2 \in \mathcal{V}} \log\left(\text{freq}(v_1, v_2) + \epsilon / \text{freq}(v_2)\right)$, where $v_1$ and $v_2$ denote the indices of two words in the vocabulary, $\text{freq}(v_2)$ and $\text{freq}(v_1, v_2)$ denote the numbers of documents in which $v_1$ appears and $v_1$ and $v_2$ co-occur, respectively, and $\epsilon = 0.01$ is used to prevent taking log of zero. Coherence is considered well-aligned to human judgment when evaluating a single topic — a higher coherence score means better quality of a mined topic. However, coherence does not evaluate the relationship between different mined topics; e.g., if the mined $F$ topics are identical, the coherence score can still be high but meaningless. To alleviate this, we also use the *similarity count* (`SimCount`) that was adopted in [10] — for each topic, the similarity count is obtained simply by adding up the overlapped words of the topics within the leading $N$ words, and a smaller `SimCount` means the mined topics are more distinguishable. When the topics are very correlated (but different), the leading words of the topics may overlap with each other, and thus using `SimCount` might still not be enough to evaluate the results. We also include clustering accuracy (`ClustAcc`), obtained by using the mined $C_\star$ matrix to estimate the weights $\boldsymbol{W}$ of the documents, and applying $k$-means to $\boldsymbol{W}$. Since the ground-truth labels of TDT2 and Reuters are known, clustering accuracy can be calculated, and it serves as a good indicator of topic mining results.

Table 1 shows the experiment results on the TDT2 corpus. From $F = 3$ to 25, the proposed algorithm (AnchorFree) gives very promising results: for the three considered metrics, AnchorFree consistently gives better results compared to the baselines. Particularly, the `ClustAcc`'s obtained by AnchorFree are at least 30% higher compared to the baselines for all cases. In addition, the single-topic quality of the topics mined by AnchorFree is the highest in terms of coherence scores; the overlaps between topics are the smallest except for $F = 20$ and 25.

Table 2 shows the results on the Reuters-21578 corpus. In this experiment, we can see that XRAY is best in terms of single-topic quality, while AnchorFree is second best when $F > 6$. For `SimCount`, AnchorFree gives the lowest values when $F > 6$. In terms of clustering accuracy, the topics obtained by AnchorFree again lead to much higher clustering accuracies in all cases.

In terms of the runtime performance, one can see from Fig. 2(a) that FastAnchor, SNPA, XRAY and AnchorFree perform similarly on the TDT2 dataset. SPA is the fastest algorithm since it has a recursive update [6]. The SNPA and XRAY both perform nonnegative least squares-based deflation, which is computationally heavy when the vocabulary size is large, as mentioned in Remark 1. AnchorFree uses AO and small-scale linear programming, which is conceptually more difficult compared to SNPA and XRAY. However, since the linear programs involved only have $F$ variables and the number of AO iterations is usually small (smaller than 5 in practice), the runtime performance is quite satisfactory

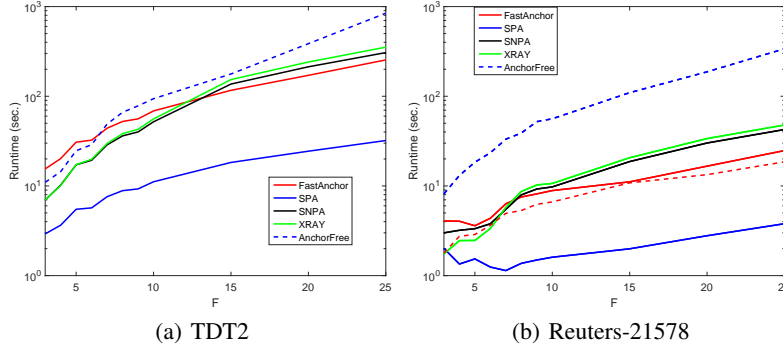

(a) TDT2

(b) Reuters-21578

Figure 2: Runtime performance of the algorithms under various settings.

Table 3: Twenty leading words of mined topics from an $F = 5$ case of the TDT2 experiment.

| FastAnchor | | | | | AnchorFree | | | | |
|---|---|---|---|---|---|---|---|---|---|
| | | anchor | | | | | anchor | | |
| predicts | slipping | cleansing | strangled | tenday | | | | | |
| allegations | poll | columbia | gm | bulls | lewinsky | gm | shuttle | bulls | jonesboro |
| lewinsky | cnnusa | shuttle | motors | jazz | monica | motors | space | jazz | arkansas |
| clinton | gallup | space | plants | nba | starr | plants | columbia | nba | school |
| lady | allegations | crew | workers | utah | grand | flint | astronauts | chicago | shooting |
| white | clinton | astronauts | michigan | finals | white | workers | nasa | game | boys |
| hillary | presidents | nasa | flint | game | jury | michigan | crew | utah | teacher |
| monica | rating | experiments | strikes | chicago | house | auto | experiments | finals | students |
| starr | lewinsky | mission | auto | jordan | clinton | plant | rats | jordan | westside |
| house | president | stories | plant | series | counsel | strikes | mission | malone | middle |
| husband | approval | fix | strike | malone | intern | gms | nervous | michael | 11year |
| dissipate | starr | repair | gms | michael | independent | strike | brain | series | fire |
| president | white | rats | idled | championship | president | union | aboard | championship | girls |
| intern | monica | unit | production | tonight | investigation | idled | system | karl | mitchell |
| affair | house | aboard | walkouts | lakers | affair | assembly | weightlessness | pippen | shootings |
| infidelity | hurting | brain | north | win | lewinskys | production | earth | basketball | suspects |
| grand | slipping | system | union | karl | relationship | north | mice | win | funerals |
| jury | americans | broken | assembly | lewinsky | sexual | shut | animals | night | children |
| sexual | public | nervous | talks | games | ken | talks | fish | sixth | killed |
| justice | sexual | cleansing | shut | basketball | former | autoworkers | neurological | games | 13year |
| obstruction | affair | dioxide | striking | night | starrs | walkouts | seven | title | johnson |

and is close to those of SNPA and XRAY which are greedy algorithms. The runtime performance on the Reuters dataset is shown in Fig. 2(b), where one can see that the deflation-based methods are faster. The reason is that the vocabulary size of the Reuters corpus is much smaller compared to that of the TDT2 corpus (18,933 v.s. 36,771).

Table 3 shows the leading words of the mined topics by FastAnchor and AnchorFree from an $F = 5$ case using the TDT2 corpus. We only present the result of FastAnchor since it gives qualitatively the best benchmark – the complete result given by all baselines can be found in the supplementary material. We see that the topics given by AnchorFree show clear diversity: Lewinsky scandal, General Motors strike, Space Shuttle Columbia, 1997 NBA finals, and a school shooting in Jonesboro, Arkansas. FastAnchor, on the other hand, exhibit great overlap on the first and the second mined topics. Lewinsky also shows up in the fifth topic mined by FastAnchor, which is mainly about the 1997 NBA finals. This showcases the clear advantage of our proposed criterion in terms of giving more meaningful and interpretable results, compared to the anchor-word based approaches.

## 5 Conclusion

In this paper, we considered identifiable anchor-free correlated topic modeling. A topic estimation criterion based on the word-word co-occurrence/correlation matrix was proposed and its identifiability conditions were proven. The proposed approach features topic identifiability guarantee under much milder conditions compared to the anchor-word assumption, and thus exhibits better robustness to model mismatch. A simple procedure that only involves one eigen-decomposition and a few small linear programs was proposed to deal with the formulated criterion. Experiments on real text corpus data showcased the effectiveness of the proposed approach.

## Acknowledgment

This work is supported in part by the National Science Foundation (NSF) under the project numbers NSF-ECCS 1608961 and NSF IIS-1247632 and in part by the Digital Technology Initiative (DTI) Seed Grant, University of Minnesota.

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
