[Supplementary Material · TMnips2016supplementary.pdf]

Supplementary Material for:
## Anchor-Free Correlated Topic Modeling:
## Identifiability and Algorithm

# 1 Proof of Proposition 1

Let us denote a feasible solution of Problem (3) in the manuscript as $(\tilde{\boldsymbol{C}}, \tilde{\boldsymbol{E}})$, and let $\boldsymbol{C}_\natural$ and $\boldsymbol{E}_\natural$ stand for the ground-truth word-topic PMF matrix and the topic correlation matrix, respectively. Note that we can represent any feasible solution as $\tilde{\boldsymbol{C}} = \boldsymbol{C}_\natural \boldsymbol{A}$, $\tilde{\boldsymbol{E}} = \boldsymbol{A}^{-1} \boldsymbol{C}_\natural \boldsymbol{A}^{-1}$ where $\boldsymbol{A} \in \mathbb{R}^{F \times F}$ is an invertible matrix. Given $\mathrm{rank}(\boldsymbol{P}) = F$ and that Assumption 1 holds, we must have

$$\mathrm{rank}(\tilde{\boldsymbol{C}}) = \mathrm{rank}(\tilde{\boldsymbol{E}}) = F,$$

for any solution pair $(\tilde{\boldsymbol{C}}, \tilde{\boldsymbol{E}})$. In fact, if the anchor-word assumption holds, then there is a nonsingular diagonal submatrix in $\boldsymbol{C}_\natural$, so $\mathrm{rank}(\boldsymbol{C}_\natural) = F$, and the same holds for $\tilde{\boldsymbol{C}} = \boldsymbol{C}_\natural \boldsymbol{A}$ since $\boldsymbol{A}$ is invertible. By the assumption $\mathrm{rank}(\boldsymbol{P}) = F$ and the equality $\boldsymbol{P} = \boldsymbol{C}_\natural \boldsymbol{E}_\natural \boldsymbol{C}_\natural^T = \tilde{\boldsymbol{C}} \tilde{\boldsymbol{E}} \tilde{\boldsymbol{C}}^T$, one can see that all the factors must have full column rank. Therefore, $|\det \tilde{\boldsymbol{E}}| > 0$ for any feasible $\tilde{\boldsymbol{E}}$ – a trivial solution cannot arise under the model considered.

Furthermore, $\tilde{\boldsymbol{C}}$ satisfies $\tilde{\boldsymbol{C}}^T \mathbf{1} = \mathbf{1}$ and $\tilde{\boldsymbol{C}} \geq \mathbf{0}$ since $\tilde{\boldsymbol{C}}$ is a solution to Problem (3). Because the rows of $\mathrm{Diag}(\boldsymbol{c})$ all appear in the rows of $\boldsymbol{C}$ under Assumption 1, a matrix $\boldsymbol{A}$ satisfies $\tilde{\boldsymbol{C}}(\Lambda, :) = \boldsymbol{C}(\Lambda, :)\boldsymbol{A} \geq \mathbf{0}$ if and only if $\boldsymbol{A} \geq \mathbf{0}$. Also note that $\boldsymbol{A}^T \boldsymbol{C}^T \mathbf{1} = \mathbf{1} \Rightarrow \boldsymbol{A}^T \mathbf{1} = \mathbf{1}$. Then, we have that

$$|\det \boldsymbol{A}| \leq \prod_{f=1}^{F} \|\boldsymbol{A}(:, f)\|_2 \leq \prod_{f=1}^{F} \|\boldsymbol{A}(:, f)\|_1 = \prod_{f=1}^{F} \boldsymbol{A}(:, f)^T \mathbf{1} = 1, \tag{1.1}$$

where the first bounding step is the Hadamard inequality, the second comes from elementary properties of vector norms, and for non-negative vectors the $\ell_1$ norm is simply the sum of all elements. The first inequality becomes equality if and only if $\boldsymbol{A}$ is a column-orthogonal matrix, and the second holds with equality if and only if $\boldsymbol{A}(:, f)$ for $f = 1, \ldots, F$ are unit vectors. Therefore, for non-negative matrices the equalities in (1.1) hold if and only if $\boldsymbol{A}$ is a permutation matrix. As a result, any alternative solution $\tilde{\boldsymbol{E}}$ has the form that $\tilde{\boldsymbol{E}} = \boldsymbol{A}^{-1} \boldsymbol{E}_\natural \boldsymbol{A}^{-1}$, and we simply have that

$$|\det \tilde{\boldsymbol{E}}| = |\det \boldsymbol{A}^{-1} \det \boldsymbol{E}_\natural \det \boldsymbol{A}^{-1}| = |\det \boldsymbol{E}||\det \boldsymbol{A}|^{-2} \geq |\det \boldsymbol{E}_\natural|,$$

where equality holds if and only if $\boldsymbol{A}$ is a permutation matrix. This means that for optimal solutions that satisfy $\boldsymbol{P} = \boldsymbol{C}_\star \boldsymbol{E}_\star \boldsymbol{C}_\star^T$, we have $\boldsymbol{C}_\star = \boldsymbol{C}_\natural \boldsymbol{\Pi}$ and $\boldsymbol{E}_\star = \boldsymbol{\Pi}^T \boldsymbol{E}_\natural \boldsymbol{\Pi}$, and achieve minimal value $|\det \boldsymbol{E}_\star|$, where $\boldsymbol{\Pi}$ is a permutation matrix. **Q.E.D.**

## 2  Proof of Lemma 1

If $\boldsymbol{C}$ is sufficiently scattered, it satisfies

$$\text{cone}(\boldsymbol{C}^T)^* \subseteq \mathcal{K}. \tag{2.1}$$

Suppose that $\boldsymbol{C}$ is rank-deficient. Then, all the vectors that lie in the null space of $\boldsymbol{C}$ satisfy $\boldsymbol{C}\boldsymbol{x} = \boldsymbol{0}$, which implies that for $\boldsymbol{x} \in \mathcal{N}(\boldsymbol{C})$ we have

$$\boldsymbol{C}\boldsymbol{x} \geq \boldsymbol{0}. \tag{2.2}$$

Eq. (2.2) and Eq. (2.1) together imply that

$$\mathcal{N}(\boldsymbol{C}) \subseteq \mathcal{K}.$$

However, a null space cannot be contained in a second-order cone, which is a contradiction.

We now show that any feasible solution pair $(\tilde{\boldsymbol{E}}, \tilde{\boldsymbol{C}})$ has full rank. Denote the ground-truth word-topic PMF matrix as $\boldsymbol{C}_\natural$, and the correlation matrix between topics as $\boldsymbol{E}_\natural$. Under Assumption 2, the ground-truth $\boldsymbol{C}_\natural$ has full column rank, and thus $\boldsymbol{E}_\natural \in \mathbb{R}^{F \times F}$ has full rank when $\text{rank}(\boldsymbol{P}) = F$. Now, since any other feasible solution can be written as $\boldsymbol{C} = \boldsymbol{C}_\natural \boldsymbol{A}$, $\boldsymbol{E} = \boldsymbol{A}^{-1}\boldsymbol{E}_\natural \boldsymbol{A}^{-1}$, where $\boldsymbol{A}$ is invertible, we have that any feasible solution pair $(\tilde{\boldsymbol{E}}, \tilde{\boldsymbol{C}})$ has full rank and $\det \tilde{\boldsymbol{E}}$ is bounded away from zero. **Q.E.D.**

## 3  Proof of Theorem 1

Denote the ground truth word-topic PMF matrix as $\boldsymbol{C}_\natural$, and the correlation matrix between topics as $\boldsymbol{E}_\natural$. What we observe is their product

$$\boldsymbol{P} = \boldsymbol{C}_\natural \boldsymbol{E}_\natural \boldsymbol{C}_\natural^T,$$

and we want to infer, from the observation $\boldsymbol{P}$, what the matrices $\boldsymbol{C}_\natural$ and $\boldsymbol{E}_\natural$ are. The method proposed in this paper is via solving (3), repeated here

$$
\begin{aligned}
&\underset{\boldsymbol{E},\boldsymbol{C}}{\text{minimize}} \ \ |\det \boldsymbol{E}| \\
&\text{subject to } \boldsymbol{P} = \boldsymbol{C}\boldsymbol{E}\boldsymbol{C}^T \\
&\qquad\qquad \boldsymbol{C}^T \boldsymbol{1} = \boldsymbol{1}, \boldsymbol{C} \geq \boldsymbol{0}.
\end{aligned}
$$

Now, denote one optimal solution of the above as $\boldsymbol{C}_\star$ and $\boldsymbol{E}_\star$, and Theorem 1 claims that if $\boldsymbol{C}_\natural$ is sufficiently scattered (cf. Assumption 2), then there exists a permutation matrix $\boldsymbol{\Pi}$ such that

$$\boldsymbol{C}_\star = \boldsymbol{C}_\natural \boldsymbol{\Pi}, \ \ \boldsymbol{E}_\star = \boldsymbol{\Pi}^T \boldsymbol{E}_\natural \boldsymbol{\Pi}.$$

Because $\text{rank}(\boldsymbol{P}) = F$, and both $\boldsymbol{C}_\natural$ and $\boldsymbol{C}_\star$ have $F$ columns, this means $\boldsymbol{C}_\natural$ and $\boldsymbol{C}_\star$ span the same column space, therefore there exists a non-singular matrix $\boldsymbol{A}$ such that

$$\boldsymbol{C}_\star = \boldsymbol{C}_\natural \boldsymbol{A}, \ \ \boldsymbol{E}_\star = \boldsymbol{A}^{-1}\boldsymbol{E}_\natural \boldsymbol{A}^{-T}.$$

In terms of problem (3), $C_\natural$ and $E_\natural$ are clearly feasible, which yields an objective value $\det E_\natural$. Since we assume $(C_\star, E_\star)$ is an optimal solution of (3), we have that

$$|\det E_\star| = |\det A^{-1} \det E_\natural \det A^{-T}|$$
$$\leq |\det E_\natural|,$$

implying

$$|\det A| \geq 1. \tag{3.1}$$

On the other hand, since $C_\star$ is feasible for (3), we also have that

$$C_\natural A \geq 0, A^T C_\natural^T \mathbf{1} = A^T \mathbf{1} = \mathbf{1}.$$

Geometrically, the inequality constraint $C_\natural A \geq 0$ means that columns of $A$ are contained in $\mathrm{cone}(C_\natural^T)^*$. We assume $C_\natural$ is sufficiently scattered, therefore

$$A(:,f) \in \mathrm{cone}(C_\natural^T)^* \subseteq \mathcal{K},$$

or equivalently

$$\|A(:,f)\|_2 \leq \mathbf{1}^T A(:,f).$$

Then for matrix $A$, we have that

$$|\det A| \leq \prod_{f=1}^{F} \|A(:,f)\|_2 \leq \prod_{f=1}^{F} \mathbf{1}^T A(:,f) = 1. \tag{3.2}$$

Combining (3.1) and (3.2), we conclude that

$$|\det A| = 1.$$

Furthermore, if (3.2) holds as an equality, we must have

$$\|A(:,f)\|_2 = \mathbf{1}^T A(:,f), \ \forall \ f = 1, ..., F,$$

which, geometrically, means that the columns of $A$ all lie on the boundary of $\mathcal{K}$. However, since $C_\natural$ is sufficiently scattered,

$$\mathrm{cone}(C_\natural^T)^* \cap \mathrm{bd}\mathcal{K} = \{\lambda e_f : \lambda \geq 0, f = 1, ..., F\},$$

so $A(:,f)$ being contained in $\mathrm{cone}(C_\natural^T)^*$ then implies that columns of $A$ can only be selected from the columns of the identity matrix $I$. Together with the fact that $A$ should be non-singular, we have that $A$ can only be a permutation matrix. **Q.E.D.**

## 4  Synthetic Experiments

In this section we give simulation results showing the word-topic PMF matrix $C$ and topic correlation matrix $E$ can indeed be exactly recovered even in the absence of anchor words, using synthetic data. For a given vocabulary size $V = 1000$ and number of topics $F$ increasing from 5 to 30, ground truth matrices $C_\natural$ and $E_\natural$ are synthetically generated: the entries of $C_\natural$ are first drawn from an

i.i.d. exponential distribution, and then approximately 50% of the entries are randomly set to zero, according to an i.i.d. Bernoulli distribution; matrix $\boldsymbol{E}_\natural$ is obtained from $\boldsymbol{R}^T \boldsymbol{R}/F$, where entries of the $F \times F$ matrix $\boldsymbol{R}$ are drawn from an i.i.d. Gaussian distribution. In this way, $\boldsymbol{C}_\natural$ is sufficiently scattered with very high probability, but is very unlikely to satisfy the separability / anchor-word assumption.

Using the synthetically generated $\boldsymbol{C}_\natural$ and $\boldsymbol{E}_\natural$, we set the word co-occurrence matrix $\boldsymbol{P} = \boldsymbol{C}_\natural \boldsymbol{E}_\natural \boldsymbol{C}_\natural^T$, and apply various topic modeling algorithms on $\boldsymbol{P}$ to try to recover $\boldsymbol{C}_\natural$ and $\boldsymbol{E}_\natural$, including our proposed AnchorFree described in Algorithm 2. Denoting the output of any algorithm as $\boldsymbol{C}_\star$ and $\boldsymbol{E}_\star$, before we compare them with the ground truth $\boldsymbol{C}_\natural$ and $\boldsymbol{E}_\natural$, we need to fix the permutation ambiguity. This task can be formulated as the following optimization problem

$$\underset{\boldsymbol{\Pi}}{\text{minimize}} \quad \|\boldsymbol{C}_\star - \boldsymbol{C}_\natural \boldsymbol{\Pi}\|_F^2$$

$$\text{subject to} \quad \boldsymbol{\Pi} \text{ is a permutation matrix}$$

which is equivalent to the *linear assignment* problem, and can be solved efficiently via the *Hungarian algorithm*. After optimally matching the columns of $\boldsymbol{C}_\star$ and $\boldsymbol{C}_\natural$, the estimation errors $\|\boldsymbol{C}_\star - \boldsymbol{C}_\natural\|_F^2$ and $\|\boldsymbol{E}_\star - \boldsymbol{E}_\natural\|_F^2$ given by different methods are shown in Table 1 and 2, where each estimation error is averaged over 10 Monte-Carlo trials. Based on the results shown in Table 1 and 2, several comments are in order:

1. The anchor-word-based algorithms are not able to recover the ground-truth $\boldsymbol{C}_\natural$ and $\boldsymbol{E}_\natural$, since the separability / anchor-word assumption is grossly violated;

2. AnchorFree, on the other hand, recovers $\boldsymbol{C}_\natural$ and $\boldsymbol{E}_\natural$ almost perfectly in all the cases under test, which supports our claim in Theorem 1;

3. Even though the identification criterion (3) is a non-convex optimization problem, the proposed procedure empirically always works, which is obviously encouraging and deserves future study.

Table 1: Estimation error $\|\boldsymbol{C}_\star - \boldsymbol{C}_\natural\|_F^2$ on synthetic data.

| $F$ | FastAnchor | SPA | SNPA | XRAY | AnchorFree |
|---|---|---|---|---|---|
| 5 | 0.0284 | 0.0284 | 0.0159 | 0.0800 | **5.83e-20** |
| 10 | 0.1984 | 0.1984 | 0.3746 | 0.0284 | **4.35e-16** |
| 15 | 0.6317 | 0.5356 | 0.7454 | 0.0509 | **8.44e-10** |
| 20 | 0.2610 | 0.2261 | 0.1776 | 0.0698 | **1.00e-09** |
| 25 | 0.2185 | 0.2235 | 0.1758 | 0.1228 | **6.67e-15** |
| 30 | 0.2999 | 0.2769 | 0.2927 | 0.1471 | **2.65e-15** |

# 5   Complete Results of the Illustrative Example

The complete results of the illustrative example in the manuscript are presented in Tables 3-7. One observation is that FastAnchor, SPA and SNPA give the same anchor words and topics with different orders. XRAY gives different anchor words and topics, but the topics mined are qualitatively worse compared to those of FastAnchor, SPA and SNPA. The proposed AnchorFree algorithm yields five clean topics.

Table 2: Estimation error $\|\boldsymbol{E}_\star - \boldsymbol{E}_\natural\|_F^2$ on synthetic data.

| $F$ | FastAnchor | SPA | SNPA | XRAY | AnchorFree |
|---|---|---|---|---|---|
| 5 | 2.08e10 | 2.08e10 | 1.00e10 | 8.79 | **1.33e-16** |
| 10 | 7.73e5 | 1.75e7 | 1.75e7 | 13.46 | **1.04e-12** |
| 15 | 1.90e6 | 2.86e6 | 3.39e6 | 34.17 | **1.91e-06** |
| 20 | 1.92e5 | 6.44e5 | 2.37e5 | 30.17 | **9.46e-07** |
| 25 | 7.97e4 | 1.09e4 | 2.15e4 | 40.25 | **1.34e-11** |
| 30 | 1.03e5 | 1.03e4 | 1.12e4 | 67.54 | **5.80e-12** |

Table 3: TDT2; $F = 5$.

| AnchorFree | | | | |
|---|---|---|---|---|
| anchor | | | | |
| lewinsky | gm | shuttle | bulls | jonesboro |
| monica | motors | space | jazz | arkansas |
| starr | plants | columbia | nba | school |
| grand | flint | astronauts | chicago | shooting |
| white | workers | nasa | game | boys |
| jury | michigan | crew | utah | teacher |
| house | auto | experiments | finals | students |
| clinton | plant | rats | jordan | westside |
| counsel | strikes | mission | malone | middle |
| intern | gms | nervous | michael | 11year |
| independent | strike | brain | series | fire |
| president | union | aboard | championship | girls |
| investigation | idled | system | karl | mitchell |
| affair | assembly | weightlessness | pippen | shootings |
| lewinskys | production | earth | basketball | suspects |
| relationship | north | mice | win | funerals |
| sexual | shut | animals | night | children |
| ken | talks | fish | sixth | killed |
| former | autoworkers | neurological | games | 13year |
| starrs | walkouts | seven | title | johnson |

Table 4: TDT2; $F = 5$.

| FastAnchor | | | | |
|---|---|---|---|---|
| | | anchor | | |
| predicts | slipping | cleansing | strangled | tenday |
| allegations | poll | columbia | gm | bulls |
| lewinsky | cnnusa | shuttle | motors | jazz |
| clinton | gallup | space | plants | nba |
| lady | allegations | crew | workers | utah |
| white | clinton | astronauts | michigan | finals |
| hillary | presidents | nasa | flint | game |
| monica | rating | experiments | strikes | chicago |
| starr | lewinsky | mission | auto | jordan |
| house | president | stories | plant | series |
| husband | approval | fix | strike | malone |
| dissipate | starr | repair | gms | michael |
| president | white | rats | idled | championship |
| intern | monica | unit | production | tonight |
| affair | house | aboard | walkouts | lakers |
| infidelity | hurting | brain | north | win |
| grand | slipping | system | union | karl |
| jury | americans | broken | assembly | lewinsky |
| sexual | public | nervous | talks | games |
| justice | sexual | cleansing | shut | basketball |
| obstruction | affair | dioxide | striking | night |

Table 5: TDT2; $F = 5$.

| XRAY | | | | |
|---|---|---|---|---|
| | | anchor | | |
| strangled | topping | dioxide | reprieve | indicting |
| gm | lewinsky | shuttle | lewinsky | lewinsky |
| motors | monica | columbia | grand | starr |
| plants | white | space | jury | monica |
| workers | starr | astronauts | monica | counsel |
| michigan | house | nasa | starr | grand |
| flint | grand | crew | white | jury |
| strikes | intern | dioxide | house | independent |
| auto | jury | experiments | counsel | ken |
| plant | clinton | mission | clinton | white |
| strike | affair | carbon | whitewater | investigation |
| gms | counsel | aboard | independent | house |
| idled | former | rats | lewinskys | clinton |
| production | independent | system | investigation | intern |
| walkouts | lie | brain | president | starrs |
| north | president | nervous | intern | stories |
| union | lewinskys | earth | ken | president |
| assembly | relationship | shut | relationship | lewinskys |
| talks | allegations | weightlessness | testimony | checking |
| shut | ken | unit | testify | former |
| striking | sexual | mice | starrs | affair |

Table 6: TDT2; $F = 5$.

| | | anchor | | |
|---|---|---|---|---|
| predicts | slipping | cleansing | strangled | tenday |
| allegations | oll | columbia | gm | bulls |
| lewinsky | cnnusa | shuttle | motors | jazz |
| clinton | gallup | space | plants | nba |
| lady | allegations | crew | workers | utah |
| white | clinton | astronauts | michigan | finals |
| hillary | presidents | nasa | flint | game |
| monica | rating | experiments | strikes | chicago |
| starr | lewinsky | mission | auto | jordan |
| house | president | stories | plant | series |
| husband | approval | fix | strike | malone |
| dissipate | starr | repair | gms | michael |
| president | white | rats | idled | championship |
| intern | monica | unit | production | tonight |
| affair | house | aboard | walkouts | lakers |
| infidelity | hurting | brain | north | win |
| grand | slipping | system | union | karl |
| jury | americans | broken | assembly | lewinsky |
| sexual | public | nervous | talks | games |
| justice | sexual | cleansing | shut | basketball |
| obstruction | affair | dioxide | striking | night |

Table 7: TDT2; $F = 5$.

| | | anchor | | |
|---|---|---|---|---|
| slipping | predicts | tenday | cleansing | strangled |
| poll | allegations | bulls | columbia | gm |
| cnnusa | lewinsky | jazz | shuttle | motors |
| gallup | clinton | nba | space | plants |
| allegations | lady | utah | crew | workers |
| clinton | white | finals | astronauts | michigan |
| presidents | hillary | game | nasa | flint |
| rating | monica | chicago | experiments | strikes |
| lewinsky | starr | jordan | mission | auto |
| president | house | series | stories | plant |
| approval | husband | malone | fix | strike |
| starr | dissipate | michael | repair | gms |
| white | president | championship | rats | idled |
| monica | intern | tonight | unit | production |
| house | affair | lakers | aboard | walkouts |
| hurting | infidelity | win | brain | north |
| slipping | grand | karl | system | union |
| americans | jury | lewinsky | broken | assembly |
| public | sexual | games | nervous | talks |
| sexual | justice | basketball | cleansing | shut |
| affair | obstruction | night | dioxide | striking |