[Reviews · NeurIPS 2016]

Reviewer 1

Summary

This paper presents a provably correct method for identifying topics in documents that moves beyond the "anchor-free" assumption in existing work.

Qualitative Assessment

[update after author response]: the authors' explanation for decreasing accuracy as F increases makes perfect sense. If it was in the paper that F is set equal to the number of classes, I missed it. Please consider adding a sentence giving your explanation for the decrease, in any case. I also appreciate the authors reporting the accuracy of a simple LDA baseline in the author response. In general the author response addressed my major concerns. [END update after author response] In general I think this paper is a strong contribution. I discuss particular strengths and areas for improvement below. Strengths: The work gives a technique with novel provable guarantees. It also provides an experimental evaluation of the approach. Parts of the paper are brilliantly clear. Figure 1 is extremely helpful for understanding the new power of the technique. The math is succinct and clear. Areas for Improvement: Lines 209-218 introduce some questions about the novelty of the underlying approach, even if this paper's algorithm itself is novel. The scalability of the technique is not great. This is shown in both the empirical results (Fig 2) and also the complexity O(F^3.5). However, this is not a fatal limitation. The authors should be aware that F values in the several thousands often provide the best held-out likelihood in practice, for even moderate-sized corpora, especially when topic correlations are modeled. The experiments could use some enhancements. First, a vanilla LDA approach using Gibbs sampling is worth including as a simple baseline. Second, there is a major unanswered question in how clustering accuracy (which in my view is the most important experimental metric, because it reflects best what users want out of a topic model) decreases with the number of topics. Especially when the number of classes considered (for TDT2 at least) is larger than the number of topics, I do not understand why as F increases, classification accuracy goes from outstanding (0.98 accuracy for F=3, TDT2) to mediocre (0.74 for F=25). The 0.98 is hard to understand by itself, when each document vector is only length 3 and you are performing a 30-way classification task. Some explanation is necessary here. Also, another approach that used the W vectors as feature vectors, and trained a supervised classifier on a small number of labeled documents, might yield more meaningful results. This experimental detail was the primary reason I did not assign a higher score. The background section could be reduced considerably in order to provide more space for experiments. Minor: The point about how normalization can increase noise should be clarified with an example, or more explanation. The paper returns to this claim at multiple points in the paper but without reading [8] it is hard for me to understand why normalization amplifies noise. Is this because documents with few supporting observations will receive equal weight in the model as those with more observations? The phrase third or higher-order statistics (trigrams) is a little confusing, because the paper is talking about word co-occurrance within the same document, whereas trigrams refer to a more specific notion (three *consecutive* words) Line 1 in Algorithm 1 says Epsilon = 1^TD^T whereas the text has Diag of that. "the size of P does not grow with the size of the data if the vocabulary is fixed" -- only true if P is stored in a dense format, but normally wouldn't it be quite sparse? I didn't understand what it meant for topics to be "uncorrelated" in the previous work, defining this more precisely would help How many document classes were used from ModApte? The paper characterizes its algorithm's complexity as F^3.5, but won't some steps, like line 2, depend on the number of non-zero entries of P? Perhaps this is negligible with current eigen-decomposition code. Two especially minor comments on first proof in the supplementary material. It's confusing that you refer to Assumption 1 by two different names ("Assumption 1" and the "anchor-word" assumption). I also think you should unpack the sentence: "Therefore, for non-negative matrices the equalities in (1.1) hold if and only if A is a permutation matrix." Saying "because any non-negative square matrix with orthonormal columns must be a permutation matrix" instead of "therefore" might help.

Confidence in this Review

2-Confident (read it all; understood it all reasonably well)


Reviewer 2

Summary

Relaxes the popular "anchor" assumption in efficient NMF algorithms for topic modeling, and shows that the resulting algorithm performs overall better than several popular anchor-based algorithms on two moderately sized topic modeling problems.

Qualitative Assessment

The main argument is a bit dense technically, but the authors provide a good intuitive explanation in figure 1 and in the discussion of Algorithm 2 in Section 3.2. The experimental results show that Algorithm 2 gets mostly better metrics than competing anchor-based algorithms. It would have been good if the paper showed the algorithm’s behavior on synthetic data as the sufficiently scattered assumption is increasingly violated. Does it degrade gracefully, or does it crash badly? Also, how well does the assumption match the experimental data? (Added after author response: The synthetic data experiment in the supplementary material nicely shows that the algorithm works well on non-separable but sufficiently scattered conditions. It would be great to have an analogous experiment that shows how the algorithm degrades as sufficient scattering is increasingly violated. It would also be useful to look at the statistics of empirical data to see how well it matches the sufficiently scattered assumption.) Minor question: in Section 2, the authors mention P’s robustness to zero-mean white noise, but that is not the kind of noise that matters for empirical word co-occurrence counts, is it?

Confidence in this Review

1-Less confident (might not have understood significant parts)


Reviewer 3

Summary

This paper proposes a new algorithm for (correlated) topic modeling that works without the anchor-words assumption. The main idea of the algorithm is based on minimizing the determinant of a "topic-topic correlation" matrix, which is related to the idea of minimizing the volume of the simplex (but different because it works on the topic-topic matrix).

Qualitative Assessment

The main idea of the algorithm is to represent the word-word correlation matrix as CEC^T, where C is the topic-word matrix that we hope to learn, and E is a topic-topic correlation matrix. The paper then tries to minimize the determinant of the matrix E. In theory, the paper gives identifiability results on the uniqueness of the solution, which is weaker than the previous anchor-words assumption. The optimization problem is solved using an alternating minimization procedure in [9]. There are a few issues in the proposed approach: 1. The minimization problem (minimizing a determinant subject to linear inequalities) is NP-hard in general, and the alternating minimization algorithm may get stuck at local optimum. This is not discussed in the paper and the paper compares against other algorithms that do have provable guarantees. This makes the comparison a bit unfair. 2. In Remark 2 (page 5), the matrix E can be any symmetric matrix that may even have negative entries. However, for topic models this is not allowed as E should be the matrix of topic-topic correlations, and as any other correlation matrix its entries should be nonnegative. This could potentially hurt the intepretability of the result (e.g. what does it mean to say "the probability that the first word is in topic i and second word is in topic j is -0.1"?). The paper claims this captures negative correlation but negative correlation is really a different thing (say two events X and Y are negatively correlated, it just means P[X and Y] <= P[X]P[Y], not P[X and Y] < 0). (Since the authors promised to further clarify 1 and 2 in the final version I'm changing the technical quality score) 3. In the experimentes it's not clear the algorithm is strongly favored against all previous algorithms, for example XRAY seems to be better in the coherent metric for Reuters, and the only metric where the new algorithm dominates is the clustering accuracy. The comparisons are also in the case when number of topics is fairly small, where the FastAnchor algorithm is known to fail (see improvement from "Robust Spectral Inference for Joint Stochastic Matrix Factorization" by Moontae Lee, David Bindel and David Mimno). So the experiment results are not very convincing. Overall I feel the idea seems interesting but the paper may still need improvement.

Confidence in this Review

2-Confident (read it all; understood it all reasonably well)


Reviewer 4

Summary

Describes a novel optimization problem, and associated algorithm, for topic modeling (non-negative matrix factorization). Gives results on a number of benchmarks.

Qualitative Assessment

I found this an interesting paper: plenty of interesting insights, a novel algorithm, good results for the method. One relatively major reservation that I have: the authors end up with an optimization problem that is non-convex, and if I understand things correctly, the algorithm is not guaranteed to solve the problem exactly(*)? In addition, there is no sample complexity result given. This differs substantially from the work of Arora et al. for example, where the whole idea was to derive an algorithm with provable guarantees. This puts the method in a different category from much of the other work that is cited. An obvious question then is whether other algorithms without provable (PAC-style) guarantees - for example EM or Gibbs sampling - would also perform well on the given benchmarks. * Actually this point really needs to be clarified in the paper, it's not clear to me whether the authors are claiming that their method has guarantees of solving the underlying problem. Update: thanks to the authors for their response to the reviews, it does help to clarify my questions above. I still think that the comparison to EM/Gibbs sampling needs to be discussed in more detail in the paper, but that there are enough interesting ideas in this paper to keep my rating as borderline leaning towards accept.

Confidence in this Review

2-Confident (read it all; understood it all reasonably well)


Reviewer 5

Summary

This paper studies the identifiable conditions in topic modeling using second-order word-word co-occurrence statistics and proposed conditions that are mild than the "anchor-word" assumptions in prior works such us [9][10]. The paper demonstrates that the proposed approach can achieve consistent topic estimation with provable guarantees under much milder conditions compared to anchor-word assumption. The paper also considers a set of experiments to show that the proposed algorithm can empirically achieve better classification accuracy in document classification application.

Qualitative Assessment

(on prior art) The paper conducted a detailed comparison against anchor-word based approaches and categorized the main issues of SPA-based approach and linear-programming based approaches. I would like to note that there are some other works in the same domain that tries to avoid these computation issues. For example, Bansal et al., NIPS14 proposed a t-SVD based approach that can relax the anchor-word assumption and is computationally robust (non-sequential); Ge and Zhou, ICML15 also considered some general conditions other than anchor-word assumptions. In addition, Lee et al., NIPS15 considers a symmetric NMF on word-word co-occurrence matrix. I think the claims of this paper would be more convincing if these old/recent works on anchor-based assumptions are taken into consideration. (line 151-153) I wouldn't agree with the argument on the uniqueness of P=CEC. Noting that C is non-negative, E is non-negative and positive semi-definite, so \widetilde(C) and \widetilde(E) may not be valid solution. (line 168-171) The condition proposed in Assumption 2 looks very similar to that in Ge and Zhou, ICML 2015. The geometric interpretation is also quite similar. It would be more convincing if the paper would compare against this work and demonstrate differences and similarities. (on algorithm) The algorithm proposed is essentially non-convex. Therefore, it is not clear if the propose algorithm can converge to the global minimum with polynomial time complexity. I think this is important to clarify in the introduction that this paper is not about consistent algorithms but identifiability. (on experiments) - it is unclear how W (topic-document mixture matrix) is estimated given \hat(C) and documents. (line 282-284) In addition, it would be more convincing to use standard classification tools rather than K-means clustering in prediction document class labels. As in typical topic modeling literature, increasing the number of topics (F) will improve the classification accuracy using off-the-shelf algorithms such as logistic regression or SVM. The results in table 1 and 2, however, indicates that as the number of topics increase, the accuracy decreases. - it would be nice if the author also include standard topic modeling algorithms such as MCMC or variational approximation in the experimental validation. These methods, although come with no guarantees, typically achieves the best empirical performances. These baseline could somehow indicate the performance of the proposed methods.

Confidence in this Review

3-Expert (read the paper in detail, know the area, quite certain of my opinion)


Reviewer 6

Summary

The second or third-order moment methods have been popularized recently in topic modeling due to their noise robustness and power to combine method of moments to likelihood-based training. This paper argues that the core assumption: separability provides identifiability but too strong to be just identifiable. Instead, the authors show that sufficient scattering of pivotal words are enough for providing the unique solution up to the trivial topic permutations. They formulate the problem as a pure optimization without the separability assumption, demonstrating that the proposed algorithm can find better topics than several baselines.

Qualitative Assessment

(Major comments:) Anchor-word algorithm is not a mere 2nd-order factorization approach, but a subtle combination of spectral, statistical, and probabilistic methods. Arora et al published two papers [9] and [10]. [9] was first proposed based mostly on spectral and statistical structures. While it contained many provable guarantees, the method was generally impractical due to the run-time costs from solving a number of LPs and uninterpretable results with many negative entries. Later [10] incorporated probabilistic structure based on the graphical model of LDA, eliminating negative entries in word-topic matrix (except topic-topic matrix) by utilizing probabilistic inference. [10] also changed the way to generate co-occurrence matrix drastically from [9], which became an statistically unbiased estimator based on underlying distributional assumption. However, it was later shown in another paper that such construction loses geometrical properties such as positive semi-definiteness, which was hold in [9] and crucial to its performance. The paper shows that [10] produces irrelevant anchor-words at the beginning, demonstrating that the fast anchor word algorithm does not extract good topics in various real dataset, if the number of topics is smaller than 25. The potentially serious problem of this paper is that the authors try to construct co-occurrence matrix based on the style of [9] (even more deviated by starting from the tf-idf word-document matrix) and then run every experiment based on the formalism of [10] where unbiased construction and probabilistic structures are essential. Moreover, the experimental results are focused only on very small number of topics where [10] was proven not to work well. Thus it is unclear whether the baselines produce their best results. Whereas evaluating topic model could be relatively subjective, in addition, there are many standard metrics and tests such as held-out likelihood, specificity and word-intrusion test which can jointly verify the effectiveness of the proposed methods. Lastly, it is unable to find any useful argument toward the quality of topic-topic matrix, which was one of the important revisions from [10] to the follow-up work. Because the objective function explicitly includes the determinant of E and the paper argues its ability to find the correlation between topics in both title and introduction, having more experiments based on E will fully complete the paper. (Minor comments:) There are many missing important details which could be crucial in understanding. Especially, experimental procedures seem highly abbreviated, possibly making readers confused. Mathematical symbols and definitions can also be further improved by changing wordings and explanations. For example, the points in non-negative orthant trivially belong to the script K. Similarly, Figure 1 can be better understood if it includes at least one 3D figure from which subfigures are sliced. Besides, there are several important but missing relevant work for anchor-word algorithm, which either tries to relax anchor-word assumptions or solve various existing issues in [10] Q1: Assuming normalization divides each entry by large count numbers, how exactly does the normalization increase noises? Q2: RecoverL2 eventually needs full and row-normalization. Have you run without any normalization? Q3: Any qualitative or qunatitative loss from removing probabilistic structures from [10]? (e.g., interpretability)

Confidence in this Review

2-Confident (read it all; understood it all reasonably well)